# Statistical Copolymers of N-Vinylpyrrolidone and Isobornyl Methacrylate via Free Radical and RAFT Polymerization: Monomer Reactivity Ratios, Thermal Properties, and Kinetics of Thermal Decomposition

**DOI:** 10.3390/polym13050778

**Published:** 2021-03-03

**Authors:** Olga Kokkorogianni, Philippos Kontoes-Georgoudakis, Maria Athanasopoulou, Nikolaos Polizos, Marinos Pitsikalis

**Affiliations:** Industrial Chemistry Laboratory, Department of Chemistry, National and Kapodistrian University of Athens, Panepistimiopolis Zografou, 15771 Athens, Greece; olga.kokkorogianni@gmail.com (O.K.); kontoes10@hotmail.com (P.K.-G.); athanamaria@gmail.com (M.A.); nikospolyz93@yahoo.g (N.P.)

**Keywords:** statistical copolymers, RAFT, N-vinylpyrrolidone, isobornyl methacrylate, reactivity ratios, thermal properties

## Abstract

The synthesis of statistical copolymers of N-vinylpyrrolidone (NVP) with isobornyl methacrylate (IBMA) was conducted by free radical and reversible addition-fragmentation chain transfer (RAFT) polymerization. The reactivity ratios were estimated using the Finemann-Ross, inverted Fineman-Ross, Kelen-Tüdos, extended Kelen-Tüdos and Barson-Fenn graphical methods, along with the computer program COPOINT, modified to both the terminal and the penultimate models. According to COPOINT the reactivity ratios were found to be equal to 0.292 for NVP and 2.673 for IBMA for conventional radical polymerization, whereas for RAFT polymerization and for the penultimate model the following reactivity ratios were obtained: r_11_ = 4.466, r_22_ = 0, r_21_ = 14.830, and r_12_ = 0 (1 stands for NVP and 2 for IBMA). In all cases, the NVP reactivity ratio was significantly lower than that of IBMA. Structural parameters of the copolymers were obtained by calculating the dyad sequence fractions and the mean sequence length. The thermal properties of the copolymers were studied by differential scanning calorimetry (DSC), thermogravimetric analysis (TGA), and differential thermogravimetry (DTG). The results were compared with those of the respective homopolymers.

## 1. Introduction

Conventional radical polymerization is undoubtedly the most effective technique to produce polymers on an industrial scale [1]. The success of this process is attributed to the following factors: foremost there is no need for complex purification methods of monomers, solvents, etc., since free radical polymerizations require just the absence of oxygen in order to be successful, it provides flexible experimental conditions and can be applied to an ample range of monomers, solvents, and temperature scales. Among other advantages is the low cost of this method, which is probably the most important reason for its commercial success, since it is applied for the synthesis of nearly 50% of all industrial polymeric materials.

Another crucial advantage of free radical polymerization is its use for the synthesis of statistical copolymers, where the most important limitation of radical polymerization, the absence of control, is an impediment. Radical polymerization fails to be considered as a “living” polymerization technique. Anionic [2,3,4], cationic [5], group transfer [6], and ring opening metathesis [7] (ROMP) polymerization are known as the most efficient “living” polymerization techniques, but their demanding and costly conditions along with the rather restricted application to certain monomers often limit their industrial exploitation. These disadvantages led to the formation of several approaches to controlling radical polymerization [8], the most recent and promising of them being the reversible addition fragmentation chain transfer (RAFT) process [9,10,11].

The RAFT process retains the main advantages of free radical polymerization, i.e., it is suitable for a great range of monomers and has high tolerance to the various functional groups, it offers non-demanding experimental conditions, but the most remarkable feature of this technique is its ability to provide polymers in a controlled manner. Although the most important and possibly tricky part of RAFT is the choice of the appropriate monomer-chain transfer agent (CTA) combination, if done carefully it can result in remarkable materials. Therefore, it can safely be accounted as a breakthrough in the field of controlled radical polymerization [12,13].

The choice of N-vinylpyrrolidone, NVP, and isobornyl methacrylate, IBMA, is of high interest. Vinyl and in particular methacrylic monomers are widely used in numerous applications, and the result of their copolymerization can be fascinating [14,15,16,17,18,19]. NVP products are of great industrial importance as they can have various applications, most importantly, in biomedical sectors and cosmetic industry. The solubility of this polymer in both aqueous and non-aqueous media, its excellent bio-tolerance is what makes this material one of the most prominent chemicals in bio-related applications [20,21,22,23,24,25,26].

Isobornyl methacrylate is a monomer that is derived from sustainable resources. As a polymeric material it has got a high glass transition temperature (Tg_IBMA_ = 110 °C–200 °C) which depends on the polymer molecular weight and tacticity. [27,28,29] The high Tg of this polymer enables its applications toward engineering thermoplastics. When copolymerized it can improve the thermal stability of resulting product and can lead to its use as a heat resistive material [30,31,32]. Another important application of the IBMA products is in fiber optics, as the polymer presents high optical transparency [33,34].

This study focuses on the statistical copolymerization of NVP and IBMA. Both conventional radical and RAFT polymerization techniques were employed to investigate the effect of the copolymerization methodology on the molecular and structural characteristics of the copolymers. NVP can only be polymerized by radical polymerization, because its amide keto group cannot be conjugated with the vinyl group. In particular, for RAFT polymerization NVP is considered to belong to the family of the less activated monomers [35,36,37,38]. These are monomers having a double bond that is connected to a saturated carbon atom or is conjugated to a lone pair on oxygen or nitrogen, as in the case of NVP. Polymerization of these monomers produces poorly stabilized radicals. On the other hand, methacrylates belong to the family of the more activated monomers in RAFT polymerization. In this case, the double bond is conjugated to an unsaturated system, such as nitrile, aromatic ring, or carbonyl groups (e.g., the methacrylate monomers). Highly stabilized radicals are produced form these monomers, due to extended resonance effects. Therefore, a considerable difference of reactivity is expected between NVP and IBMA in the present study.

The reactivity ratios are routinely employed to express the kinetic preference over the copolymerization reaction of two monomers [39]. In the present study, they were explicitly calculated by various approaches, including both graphical (such as Finemann-Ross, inverted Fineman-Ross, Kelen-Tüdos, extended Kelen-Tüdos, and Barson-Fenn plots) and non-graphical methods, employing the computer program COPOINT. The monomer dyad fractions and the mean monomer sequence lengths were calculated as well. Both the terminal and the penultimate model were tested, in order to study thoroughly the copolymerization behavior. The thermal properties of the statistical copolymers were determined by differential scanning calorimetry (DSC), thermogravimetric analysis (TGA), and differential thermogravimetry (DTG) measurements.

## 2. Materials and Methods

### 2.1. Materials

N-Vinylpyrrolidone (≥97% FLUCA, Bucharest, Romania) containing sodium hydroxide as inhibitor was dried overnight over calcium hydride and was distilled prior to use. Isobornyl methacrylate (TCI Chemicals, Tokyo, Japan) stabilized with methyl hydroquinone, was dried as well over calcium hydride overnight and then passed through a MEHQ inhibitor remover column. Azobisisobutyronitrile AIBN (98% ACROS Chemicals, Gotëborg, Sweden) was purified by recrystallization twice from methanol and was then dried under vacuum. The chain transfer agent [(O-ethylxanthyl)methyl]benzene was synthesized according to the literature [40]. Chloroform-d (Acros Organics, Gotëborg, Sweden) was used as purchased. 1,4-dioxane (Fisher Chemicals, Loughborough, UK) was passed through a basic alumina (Al_2_CO_3_) column.

### 2.2. Synthesis of PNVP-co-PIBMA Statistical Copolymers via Free Radical and RAFT Polymerization

Both free radical and RAFT copolymerizations were performed in glass reactors employing high vacuum techniques. A set of five copolymers of NVP and IBMA were prepared for each technique. Samples prepared via free radical polymerization are characterized by the letter “F,” whereas those prepared via RAFT by the letter “R.” Different feed ratios were prepared in each copolymerization (monomer molar ratios NVP/IBMA: 80/20, 60/40, 50/50, 40/60, and 20/80). Different copolymers are denoted by the various feed molar ratios of the monomers. For example, sample F20/80 indicates the copolymer synthesized via free radical copolymerization, employing 20% NVP and 80% IBMA as molar feed composition.

After being loaded with the polymerization mixture, the reactor was adapted to a high- vacuum line, and the polymerization mixture underwent three freeze-thaw pump cycles in order to eliminate the oxygen from within. The reactor was then flame-sealed and placed in a preheated oil-bath at 60 °C.

The conventional radical copolymerization reactions were performed in bulk at 60 °C. The polymerization time was 40 min. In the RAFT process, 1,4-dioxane was employed as solvent, and the copolymerization time varied from 1.5 to 2 h. The goal is to keep a low conversion so the reactivity ratio studies can be as accurate as possible. The exact quantities of all reagents are included in Appendix A.

Finally, reactions were stopped by removing the reactor from the oil-bath and cooling the mixture under the flow of cold water. The reactor was then opened so as to expose the mixture to air.

The polymers prepared either via free radical or RAFT copolymerization were dissolved in CHCl_3_ and precipitated in cold methanol. This procedure was repeated three times in order to ensure the removal of any unreacted monomer residues. Afterwards, the polymers were dried overnight in a vacuum oven at 50 °C to remove any residual solvent.

### 2.3. Characterization Techniques

The molecular weight (M_w_) as well as the molecular weight distribution, Ð_M_ = M_w_/M_n,_ was determined by size exclusion chromatography, SEC, employing a modular instrument consisting of a Waters (Milford, MA, USA) model 510 pump, U6K sample injector, 401 differential refractometer, and a set of 5μ-Styragel columns with a continuous porosity range from 500 to 10^6^ Å. The carrier solvent was CHCl_3_ and the flow rate 1 mL/min. The system was calibrated using nine polystyrene standards with molecular weights in the range of 970–600,000.

The NMR measurements were carried out on a 400 MHz Bruker Avance Neo instrument (Bruker BioSpin, Rheinstetten, Germany) using chloroform-d as a solvent at 298 K.

The Tg values of the copolymers were determined by a 2910 Modulated DSC Model from TA Instruments (New Castle, DE, USA). The samples were heated under nitrogen atmosphere at a rate of 10 °C/min from −10 °C up to 220 °C. The second heating results were obtained in all cases.

The thermal stability and the kinetics of thermal decomposition of the copolymers were studied by thermogravimetric analysis (TGA) employing a Q50 TGA model from TA Instruments (New Castle, DE, USA). The samples were placed in a platinum pan and heated from ambient temperatures to 600 °C in a 60 mL/min flow of nitrogen at heating rates of 3, 5, 7, 10, 15, and 20 °C/min.

## 3. Results and Discussion

### 3.1. Statistical Copolymers of NVP and IBMA via Free Radical Polymerization

The free radical copolymerization of NVP and IBMA was carried out in bulk, for 40 min, at a temperature of 60 °C and AIBN was used as the polymerization initiator (Scheme 1). Since these two particular monomers had not been copolymerized before, pinpointing the exact experimental conditions, in order to receive the best possible results required specific care. After careful study of the experimental parameters a set of five copolymers with different feed compositions was prepared. The conversion was kept at a low level so that the copolymerization equation is valid.

The molecular characteristics of the samples were estimated by SEC using CHCl_3_ as the carrier solvent and are provided in Table 1. As it is obvious from the SEC traces given in Figure 1, the curves are quite symmetrical, indicating a relatively good control of the radical polymerization. The polydispersity of the copolymers varies from 1.98–2.10, a level which is considered as reasonable for free radical polymerization. The high molecular weights of the copolymers are to be expected taking into account the monomer and initiator concentrations.

The copolymer composition was determined by the ^1^H-NMR spectra of the copolymers. A characteristic example is given in Figure 2. The composition was calculated taking into account the signals 3 and (a + e), attributed to the IBMA and NVP monomer units, as shown in Figure 2.

### 3.2. Reactivity Ratios

The reactivity ratios, rr, of the statistical copolymers that were prepared by free radical polymerization, were determined by exploiting the following methods: Fineman-Ross [41], inverted Fineman-Ross [41], Kelen- Tüdos [42], extended Kelen- Tüdos [42] along with the computer program COPOINT [43]. All of the monomer reactivity ratios, in this case, were calculated in accordance with the terminal model [1,39].

In line with the Fineman-Ross method, the reactivity ratios of the monomer are to be calculated by the following equation:G = H r_IBMA_ − r_NVP_(1)
where G is plotted against H in every experiment, which grants a straight line of slope r_IBMA_ and intercept r_NVP_. The parameters G and H are defined as follows:G = X (Y − 1)/Y(2)
and
H = X^2^/Y(3)
with
X = M_IBMA_/M_NVP_(4)
and
Y = dM_IBMA_/dM_NVP_(5)
where, M_IBMA_ and M_NVP_ are the monomer molar compositions in feed as well as dM_IBMA_ and dM_NVP_ are the copolymer molar compositions.

The inverted Fineman-Ross method is specified by the following equation:G/H = r_BzMA_ − (1/H) r_NVP_(6)
in which the G/H plotted against 1/H yields r_NVP_ as the slope and r_IBMA_ as the intercept.

The Kelen-Tüdos equation can easily be considered as a refined Fineman-Ross which was produced by the introduction of an arbitrary constant (α) in order to ensure uniform distribution of the data, and the elimination of possible distortion from certain experimental data. The previously introduced G and H values are now modified resulting to values ξ and η. The H_min_ and H_max_ parameters, which represent the minimum and maximum values of H, are to be determined from the experimental data. The Kelen- Tüdös method can be summarized by the following equation:η = (r_IBMA_ + r_NVP_/α) ξ − r_NVP_/α(7)
where η and ξ are functions of the parameters G and H:η = G/(α + H)(8)
ξ = H/(α + H)(9)

α is a constant which is equal to (H_max_ H_min_)^1/2^. Plotting η as a function of ξ gives a straight line that yields -r_NVP_/α and r_IBMA_ as intercepts on extrapolation to ξ = 0 and ξ = 1, respectively.

In the extended Kelen-Tüdos equation the effect of the conversion is taken into consideration. The molar conversion of the two monomers respectively is defined as:ζ_B_ = W [(μ + X)/(μ + Y)](10)
ζ_A_ = (X/Y) ζ_B_.(11)
where W represents the weight conversion of the copolymerization, as μ represents the ratio of the molecular weight of IBMA to that of NVP. Then, z which is a conversion-dependent parameter is stated as:z = log(1 − ζ_A_)/log(1 − ζ_B_)(12)

Consequently, the previous parameters are to be redefined as: H = Y/z^2^, G = (Y − 1)/α, η = G/(α + H) and ξ = H/(α + H). The unique characteristic of the K-T and the ext. K-T methods is their ability to result in reactivity ratio data that are not affected in any way by arbitrary factors. The resulting copolymerization data are given in Table 2, and the associated graphical plots are given in Appendix A. The reactivity ratios of the copolymers that were obtained via free radical polymerization are outlined in Table 3.

All the aforementioned methods, used to determine the reactivity ratios of each monomer in the resulting copolymer, are graphical methods and are suitable for determining reactivity ratios at relatively low or even medium conversions. As previously stated, the Kelen-Tüdos method outweighs the Fineman-Ross method since it enables us to take into consideration possible changes in composition at high monomer conversions. Despite their ability to articulate more accurately the reactivity ratios of the monomers, the Kelen- Tüdos and extended Kelen- Tüdos methods are subsided by the limitations that impede all linear least square, LLS, methods, which leads them to produce less accurate values of reactivity ratios. As a way to treat errors from the LLS methods, Behnken [39] was the first to propose a nonlinear approach to the determination of reactivity ratios, which is now revolutionized by the use of computer programs, such as COPOINT, which is a non-linear least square difference procedure. COPOINT uses numeric integration techniques, which enables the user to apply a broad range of copolymerization equations in their differential form. The copolymerization parameters obtained through COPOINT are the product of the minimization that is applied to the sum of square differences in measured and calculated copolymer compositions. COPOINT uses the Mayo-Lewis equation to produce results according to the terminal model [39] and the Merz-Barb-Ham method for results depending on the penultimate model [39]. In the case of the copolymers that resulted from free radical polymerization, the use of the conventional terminal model, which is usually adequate to describe a binary polymerization, was a perfect match, since the plots contributed to each graphical method are linear, a fact which moreover shows that the copolymerizations follow the conventional kinetics. The terminal model takes into consideration the fact that the reactivity of the propagating polymer chain depends only on the last monomer unit of the growing chain and not on any units preceding the last one.

As can be concluded from the calculations, the reactivity ratio of IBMA is, in every case, significantly higher than that of NVP, or in other words NVP is the less reactive monomer in this radical copolymerization reaction. These results are in line with the literature, where NVP is frequently reported as a less reactive monomer in copolymerization procedures [44,45,46,47]. The IBMA/NVP monomer reactivity ratios relationship is r_IBMA_ > 1 > r_NVP_. Copolymerizations that show this particular rr relationship tend to form typical gradient copolymers while the polymerization takes place without an azeotropic point. The tendency of the IBMA to be integrated in the copolymer to a greater extent leads to the production of the aforementioned gradient or pseudo- diblock copolymers.

Following the reactivity ratio studies, the Igarashi equations [48] were exploited, in order to determine the statistical distribution of the dyad monomer sequences M_IBMA_-M_IBMA_, M_NVP_-M_NVP_, as well as M_IBMA_-M_NVP_. The equations proceed as following:(13)$$\;X=\varphi_{\mathrm{IBMA}}-\frac{2\varphi_{\mathrm{IBMA}}\left(1-\varphi_{\mathrm{IBMA}}\right)}{1+{\left[{\left(2\varphi_{\mathrm{IBMA}}-1\right)}^{2}+4r_{\mathrm{IBMA}}r_{\mathrm{NVP}}\varphi_{\mathrm{IBMA}}\left(1-\varphi_{\mathrm{IBMA}}\right)\right]}^{\frac{1}{2}}}$$
(14)$$\;Y=\left(1-\varphi_{\mathrm{IBMA}}\right)-\frac{2\varphi_{\mathrm{IBMA}}\left(1-\varphi_{\mathrm{IBMA}}\right)}{1+{\left[{\left(2\varphi_{\mathrm{IBMA}}-1\right)}^{2}+4r_{\mathrm{IBMA}}r_{\mathrm{NVP}}\varphi_{\mathrm{IBMA}}\left(1-\varphi_{\mathrm{IBMA}}\right)\right]}^{\frac{1}{2}}}$$
(15)$$\;Z=\frac{4\varphi_{\mathrm{IBMA}}\left(1-\varphi_{\mathrm{IBMA}}\right)}{1+{\left[{\left(2\varphi_{\mathrm{IBMA}}-1\right)}^{2}+4r_{\mathrm{IBMA}}r_{\mathrm{NVP}}\varphi_{\mathrm{IBMA}}\left(1-\varphi_{\mathrm{IBMA}}\right)\right]}^{\frac{1}{2}}}$$
where X, Y, and Z represent the mole fractions of the M_IBMA_-M_IBMA_, M_NVP_-M_NVP_, and M_IBMA_-M_NVP_ respectively, as φ_ΙΒΜA_ stands for the isobornyl methacrylate mole fraction in the resulting copolymer. Along with the dyad monomer sequences, the mean sequence lengths μ_IBMA_ and μ_NVP_ were computed with the help of the succeeding equations [49]:μ_IBMA_ = 1 + r_IBMA_ (M_IBMA_/M_NVP_)(16)
μ_NVP_ = 1 + r_NVP_ (M_NVP_/M_IBMA_)(17)

The conclusive data from the calculation of the dyads and the mean sequence length are provided in Table 4 and plotted in Figure 3.

### 3.3. Statistical Copolymers of NVP and IBMA via RAFT Polymerization

The RAFT copolymerization of NVP and IBMA was conducted at 60 °C in various times, depending on the monomer feed. AIBN was used as the polymerization initiator, whereas 1,4-dioxane was employed as a solvent and [(O-ethylxanthyl)methyl]benzene as the chain transfer agent. As in the free radical copolymerization of these two monomers, the experimental conditions were very carefully selected after the performance of several trial experiments, since their mutual RAFT copolymerization has never been attempted before, to our knowledge. After pinpointing the exact experimental parameters, a set of five copolymers was prepared and subjected to the same analytical techniques that were used on the samples resulting from the free radical polymerization. The molecular weight characteristics of the copolymers that were synthesized via RAFT polymerization are summarized in Table 1. The polydispersities from RAFT copolymerization are remarkably lower than those of the free radical copolymerization, thus indicating that the RAFT process is undoubtedly a controlled procedure. Characteristic SEC traces are given in Figure 1, whereas the ^1^H NMR spectrum of sample R60/40 is provided in the Appendix A.

The attempt to calculate the reactivity ratios of the copolymers using the terminal model proved fruitless since negative r_NVP_ values appeared in all the cases. These results are provided in Appendix A. A negative reactivity ratio value would also fail to predict a reasonable composition profile for the copolymers. In a case where the terminal model was proven to be inadequate, the negative reactivity ratio values hinted a penultimate unit effect, which therefore led us to exploit the penultimate model for sufficient data analysis. The penultimate model may be valid in a copolymerization process when there is a substantial difference in polarity, resonance, and steric hindrance effects [39,50]. In the specific case of the present work there is a huge difference in polarity between the two monomers employed, resonance is effective only in IBMA and in addition IBMA is a much bulkier monomer than NVP. These observations justify without any doubt the application of the penultimate model for the examined system.

In the latter model, two monomer reactivity ratios are given for each monomer, r which stands for the case where the penultimate and terminal monomer units are the same and r′ in which the penultimate and terminal monomer units are different. This leads to the conclusion that the penultimate effect evolves eight propagating species and four reactivity ratios described by the following equations:(18)$$r_{11}=r_{1}=\frac{k_{111}}{k_{112}},{\;r}_{21}=r_{1}^{\prime }=\frac{k_{211}}{k_{212}},{\;r}_{12}=r_{2}^{\prime }=\frac{k_{122}}{k_{121}},{\;r}_{22}=r_{2}=\frac{k_{222}}{k_{221}}$$

The versatile equation relating the feed to the copolymer composition when a penultimate kinetic effect is in operation is stated as effect is in operation is stated as:(19)$$x=\frac{1+\frac{{r^{\prime }}_{1}X\left(r_{1}X+1\right)}{\left({r^{\prime }}_{1}X+1\right)}}{1+\frac{{r^{\prime }}_{2}\left(r_{2}+X\right)}{X\left({r^{\prime }}_{2}+X\right)}}$$

The r_2_ and r_2_′ values can be safely predicted by linearization of the original Barson-Fenn equation [51]. The bulky size and steric hindrance, which are shown by IBMA lead to the conclusion that the reactivity of IBMA is not influenced by the preceding unit, so it was safely assumed that r_NVP_ and r_NVP_′ are equal to zero. Bearing that in mind, the original Barson-Fenn Equation (20) was modified (21) and used to determine more accurate and rational reactivity ratios.
r_2_ = Xk/x + X^2^k/xr′_2_ − X(20)
X[(k − x)/x] = −X^2^k/xr′_IBMA_ + r_IBMA_(21)
where,
(22)k=1−x
x = f_IBMA_/f_NVP_(23)
X = F_IBMA_/F_NVP_(24)

F gives the feed mole fraction whereas f represents the mole fraction in the copolymer. The plot of the left-hand side of Equation (20), [X(k − x)/x] against X^2^k/x states r_IBMA_ as the intercept and (−1/r′_IBMA_) as the slope.

The computer program COPOINT was employed, this time tailored to the penultimate model for the accurate prediction of the reactivity ratios. The results of both the Barson-Fenn methodology and the COPOINT program are listed in Table 5. The plot of the Barson-Fenn equation is displayed in Appendix A.

It is obvious that conducting the copolymerization of the same monomers with different methodologies, namely conventional free radical and RAFT, results in different copolymerization behavior, which can be described by different copolymerization models. This effect has been verified in the literature using as monomers NVP and n-hexyl methacrylate [52,53].

### 3.4. Thermal Properties

The thermal properties of the statistical copolymers prepared by RAFT were studied by DSC and TGA. Both PNVP and PIBMA homopolymers have high Tg values. Specifically, Tg = 187.1 °C for PNVP [54,55], whereas for PIBMA the Tg value varies depending on the molecular weight and the tacticity of the sample [27,28,29]. High molecular weight polymers of high syndiotacticity, as those reported in this study show Tg values up to 209 °C. The results of the statistical copolymers are given in Table 6. Only one Tg value was obtained, as it is reasonably expected due to the similarity of the Tg values of the respective homopolymers. A small increase in Tg was observed upon increasing the IBMA fraction in the copolymer structure.

TGA and DTG measurements were employed to provide information regarding the thermal stability and the kinetics of thermal decomposition of the statical copolymers. The measurements were conducted under different heating rates from 1 up to 20 °C/min. Characteristic thermograms from the TGA and DTG measurements are given in Figure 4, Figure 5 and Figure 6 for the PIBMA homopolymer along with PNVP-stat-PIBMA copolymers, whereas more data are provided in the Appendix A. Tables containing detailed data regarding the range of thermal decomposition (temperatures of initiation and completion of the thermal decomposition, temperature at the highest rate of thermal decomposition) are displayed in the Appendix A, as well, for different rates of heating and for all homopolymers and copolymers (Appendix A).

In all cases, both homopolymers and copolymers, the onset of thermal decomposition was shifted to higher temperatures upon increasing the heating rate. This effect was also observed in similar thermal degradation studies [54,55,56] and is due to the shorter heating time, which is required for a sample to reach a given temperature at the faster heating rate. DTG measurements for the PNVP homopolymer revealed a single decomposition maximum in the temperature range between 415 and 451 °C, indicating the presence of a rather simple mechanism of decomposition. This result can be attributed to the predominant depolymerization mechanism leading to the formation of monomers of the polymeric main chain, along with simultaneous reactions yielding oligomers [54,55].

A much more complex thermal degradation behavior was obtained for the PIBMA homopolymer. DTG profiles revealed a three-step degradation process. The first step is located at the temperature range between 235 and 250 °C and is the most pronounced, corresponding to about a 70% loss of weight of the sample. The second degradation step (25% loss of weight) is observed in the range 286–315 °C, whereas the last and minor degradation step (5% loss of weight) in the range 377–425 °C. These results imply the presence of a complex mechanism of thermal degradation of PIBMA, obviously involving the initial decomposition of the bulky ester group, followed by the thermal decomposition of the main polymeric backbone. Similar studies have been performed for other polymethacrylates, synthesized via RAFT polymerization. Specifically, poly(benzyl methacrylate), PBzMA, revealed a two-step thermal degradation process [55]. The first step was observed at 275–300 °C, corresponding to about 20% loss of weight, whereas the second step was observed at 340–460 °C. On the other hand, poly[2-(dimethylamino)ethyl methacrylate], PDMAEMA, showed a similar behavior [54]. The first thermal decomposition step, corresponding to a weight loss of 60%, was located in the range of 303–352 °C, whereas the second step in the range of 403–437 °C. More recent studies were performed with poly(stearyl methacrylate), PSMA, and poly(n-hexyl methacrylate), PHMA. [56] In these cases, single decomposition maxima were obtained by DTG analysis. The maxima were located at 270–330 °C for PSMA and 282–320 °C for PHMA. However, at lower rates of heating a small shoulder or even a second decomposition peak was observed at lower temperatures (in the range 180–270 °C).

It is clear that the nature of the ester group of the various polymethacrylates plays an important role in defining the thermal decomposition profile of the homopolymers, usually introducing complexity to the degradation mechanism. This mechanism usually involves the decomposition of the ester group initially, followed by the decomposition of the main chain at the later steps. Consequently, the thermal labile isobornyl group renders the PIBMA the less thermally stable polymethacrylate among those examined above. On the other hand, the aromatic side groups of PBzMA offer enhanced thermal stability to this homopolymer. The strong intra- and intermolecular interactions developed among the polar dimethylamino side groups of PDMAEMA introduce high thermal stability to this homopolymer as well. Taking these data into account it can be concluded that the thermal stability of the various homopolymers increases in the order PIBMA < PHMA ≈ PSMA < PDMAEMA < PBzMA.

Considering the thermal degradation profile of the PNVP and PIBMA homopolymers, it is reasonable to conclude that the decomposition of the corresponding statistical copolymers will also be complex as it will combine the properties of the thermally stable PNVP with the thermally sensitive PIBMA moieties. This expectation was verified observing three steps of thermal degradation in the copolymers. The main step, corresponding to about 80% loss of weight, is located in the temperature range 280–325 °C for all copolymers. This step is accompanied by two other degradation steps, one at lower and the other at higher temperature ranges. Comparing the statistical copolymers with the respective homopolymers, the following conclusions can be reached. The higher temperature decomposition peak of the copolymers is located in the range where the thermal decomposition of PNVP takes place. This temperature range increases upon increasing the NVP content of the copolymer. Therefore, it is attributed to the NVP units across the copolymeric chains. This peak has a rather small contribution to the total decomposition profile (15% copolymer weight loss), as is expected from the rather low composition of the copolymers in NVP units. The lower temperature decomposition peak of the copolymers is observed in the temperature range where the major weight loss of the PIBMA homopolymer is observed. The contribution of this peak is only 2–5% of the total copolymer mass, whereas in the case of the PIBMA this is the main degradation event (70% of the weight loss). On the other hand, the intermediate decomposition peak of the copolymers is greatly enhanced compared to the PIBMA homopolymers. This is direct evidence that the incorporation of the NVP units across the copolymer chain significantly enhances the thermal stability of the copolymers compared to the PIBMA homopolymers.

The activation energies, Ea, of the thermal decomposition procedure for both the homopolymers and the statistical copolymers were calculated using the well-established isoconversional Ozawa-Flynn-Wall (OFW) [57,58,59] along with the Kissinger methods. [60,61]

The reaction rate of the thermal decomposition reaction is expressed as a function of conversion α and temperature T as:dα/dt = f(α)k(T)(25)
where t is time, α is the conversion of the decomposition reaction, and f(α) the differential conversion function. The dependance on the temperature can be an Arrhenius equation, that is:k(T) = Ae^−Ea/RT^(26)
where A is the pre-exponential factor (min^−1^), Eα the activation energy, and R is the gas constant (8.314 J·mol^−1^·Κ^−1^). Substituting (26) to (25) affords:dα/dt = Ae^−Ea/RT^ f(α)(27)

In case the heating rate β is constant, that is:β = dT/dt(28)

Equation (3) is transformed to:dα/dT = (A/β) e^−Ea/RT^ f(α)(29)
or else:dα/f(α) = (A/β) e^−^^Ea/RT^dT(30)

Upon integrating Equation (30) the result is the following:(31)$$g\left(a\right)=\int_{0}^{a}\frac{\mathrm{da}}{f\left(a\right)}=\frac{A}{\beta }\;\int_{\mathrm{To}}^{T}e^{\frac{-\mathrm{Ea}}{\mathrm{RT}}}\mathrm{dT}=\frac{\mathrm{AEa}}{\beta R}P\left(x\right)$$
where To and T are the initial and final temperatures of the reaction, respectively. g(α) is the integral conversion function and x = Eα/RT [62,63,64,65,66,67]. As it is obvious, g(α) depends on the conversion mechanism and its mathematical model. Several algebraic expressions of functions of the most common reaction mechanisms operating in solid state reactions are given in the literature [68]. The P(x) function has no analytical solution. Therefore, several approximate expressions have been suggested. Among them is the following, which is known as the Doyle approximation [69]:P(x) = 0.0048e^(−1.0516x)^(32)

Substitution of Equations (32) and (9) to Equation (31) results the very well-known Ozawa-Flynn-Wall (OFW) [57,58,59] equations:(33)$$\mathrm{OFW}:\;\ln\beta =\ln\left[\frac{0.0048\mathrm{AEa}}{g\left(a\right)R}\right]-1.0516\frac{\mathrm{Ea}}{\mathrm{RT}}$$

This methodology belongs to the isoconversional approaches and is a “model free” method, taking into account that the conversion function f(α) is not affected by the change of the heating rate, β, for all values of *α*. Therefore, plotting lnβ versus 1/T should provide straight lines with slope directly proportional to the activation energy. Furthermore, if the determined activation energy values do not appreciably vary with various values of *α*, then a single-step degradation reaction can be concluded.

The OFW method involves measuring of the temperatures corresponding to fixed values of α from experiments at different heating rates β. The OFW method is the most useful method for the kinetic interpretation of thermogravimetric data, obtained from complex processes like the thermal degradation of polymers and can be applied without knowing the reaction order of the decomposition process.

In addition to these isoconversional methods the Kissinger method can also be applied to provide the activation energy Eα [60,61]. It is based on the equation:ln(*β*/Tp^2^) = ln(AR/Ea) + ln[n(1 − *α*_p_)^n−1^] − (Ea/RTp)(34)
where β is the heating rate of the samples, A is the pre-exponential factor (min^−1^), R is the gas constant (8.314 J·mol^−1^·Κ^−1^), Tp and *α*_p_ are the absolute temperature and the conversion at the maximum weight-loss, and n is the reaction order of the decomposition process. The Ea values can be calculated from the slope of the plots of ln(β/Tp^2^) versus 1/Tp.

Characteristic plots employing the Kissinger methodology are displayed in Figure 7 and Figure 8, whereas example plots employing the OFW methodology are given in Figure 9 and Figure 10. The activation energies calculated by the Kissinger methodology for all samples are shown in Table 7, whereas those obtained by the OFW approach in Table 8 for the PNVP-*co*-PIBMA copolymers, respectively. More plots from both graphical procedures are included in the Appendix A for the Kissinger plots and Appendix A for the OFW plots).

The Kissinger and the OFW plots for the PNVP, the PIBMA homopolymers and the copolymers are more or less linear with very high correlation coefficients in almost all cases, meaning that both methods are efficient to provide reliable results regarding the kinetics of their thermal decomposition. In a few OFW plots corresponding to very low (a = 0.1) or very high (a = 0.9) conversions there is a deviation from linearity, and the results are not consistent with the other data. This conclusion can be attributed to the fact that at the beginning or at the end of the thermal degradation the sample does not have the same decomposition behavior as a result of the presence of the end-groups of the polymer chains and the variation of comonomer composition along the copolymeric chain.

Since DTG revealed a multistep degradation profile for the PIBMA homopolymer and the copolymers, distinct Kissinger plots were obtained for each degradation step, leading to the calculation of three different values of Ea. The variation of Ea values with the conversion from the OFW plots is very small for PNVP, and in addition the results from both methodologies, Kissinger and OFW, are quite similar, thus indicating the presence of a rather simple thermal degradation mechanism. The situation is reversed in the case of the PIBMA homopolymer and the statistical copolymers, where the Ea values vary considerably with conversion, confirming the presence of a complex mechanism of thermal degradation. It is clear from the experimental findings that both the composition and most importantly the sequence of the monomer units along the copolymeric chain play an important role in defining the kinetics of the thermal decomposition of the copolymers.

In order to verify the effect of the polymerization method on the kinetics of the thermal decomposition, the activation energies for the thermal decomposition of the copolymer 50/50, prepared by free radical copolymerization, were calculated employing the Kissinger and OFW methodologies. The results are given in the Appendix A. It is obvious that, despite the small differences in composition and the sequences of the monomer units, coming from the different reactivity ratios, the polymerization technique does not greatly influence the decomposition behavior. The degradation profile between the 50/50 copolymers via free radical and RAFT copolymerization are similar and the Ea values from both methodologies are more or less the same.

## 4. Conclusions

Free radical and RAFT polymerization techniques were employed for the synthesis of statistical copolymers of N-vinylpyrrolidone (NVP) with isobornyl methacrylate (IBMA). As was expected, RAFT methodology afforded a much better control over the molecular characteristics of the copolymers. The reactivity ratios were measured using several linear and non-linear methods, including the computer program COPOINT. The terminal model seems to be appropriate for the description of the free radical copolymerization process. On the other hand, for the RAFT copolymerization the penultimate model proved valid for the description of the copolymerization reaction, due to the differences between the polarity, the resonance, and the steric effects of the NVP and the IBMA. All methods revealed IBMA reactivity ratios much larger than that of NVP, implying a tendency for pseudo- or gradient diblock synthesis. Specifically, according to COPOINT the reactivity ratios were found to be equal to 0.292 for NVP and 2.673 for IBMA for conventional radical polymerization, whereas for RAFT polymerization and for the penultimate model the following reactivity ratios were obtained: r_11_ = 4.466, r_22_ = 0, r_21_ = 14.830, and r_12_ = 0 (1 stands for NVP and 2 for IBMA). The PNVP homopolymers are thermally more stable than PIBMA. The mechanism of thermal degradation of PNVP is simple, whereas that of PIBMA much more complex. The statistical copolymers showed a similar multistep complex thermal degradation mechanism. The activation energies (Ea) were calculated by the Kissinger and Ozawa-Flynn-Wall (OFW) methodologies. Results on the kinetics of thermal decomposition were also obtained for one of the copolymers prepared via free radical copolymerization, showing that the copolymerization technique does not substantially influence the thermal degradation behavior.

## Data Availability

The data presented in this study are available in this manuscript and the Appendix A.

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
