# Peer review of "Statistical Copolymers of N-Vinylpyrrolidone and Isobornyl Methacrylate via Free Radical and RAFT Polymerization: Monomer Reactivity Ratios, Thermal Properties, and Kinetics of Thermal Decomposition"

_polymers, 2021, doi:10.3390/polym13050778_

Round 1

Reviewer 1 Report

Polymers 1122960

“Statistical Copolymers of N-Vinylpyrrolidone and Isobornyl Methacrylate via Free Radical and RAFT Polymerization: Monomer Reactivity Ratios, Thermal Properties and Kinetics of Thermal Decomposition”

Corresponding author: Prof. Marinos Pitsikalis

In this work, the authors synthesized statistical copolymers of N-vinylpyrrolidone with isobornyl methacrylate by free radical and RAFT polymerization. The reactivity ratios were estimated using several methods. Also, the thermal properties of the copolymers were studied.

In my opinion there are many stylistic details that need to be corrected throughout the manuscript before being accepted. Improve the quality of images, figures, equations, etc.

My comments and questions:

In the abstract part: it is recommended to include values of monomer reactivity ratios obtained in this work.

The introductory part should be extended and enriched, for example include information on the methods to calculate the comonomer reactivity ratios. Moreover, the NVP belong to less activated monomers. Please include this information.

In the experimental part:

The experimental conditions for the synthesis of PNVP-co-PIBMA statistical copolymers is not complete.

For example, the initial concentration of the xanthate agent and the initiator must be provided for each concentration.

Detailed information on free radical polymerization should also be provided. For example, “the polymers were dissolved in CHCl3…”.. Do the authors refer to the polymers synthesized via free radicals? Please clarify.

Line 105. The purification process of the polymers obtained must be differentiated depending on the type of polymerization.

Line 149: In the 1H NMR spectrum, hydrogens of the copolymers should be assigned in detail.

Also, a 1H NMR spectrum for R60/40 should be included. The signals of the hydrogens from the CTA will be observed in low field.

Line 241: The quality of the image in which the equations are displayed must be improved. for example: eq 13 to 15. Also eq. 18, 19.

Line 497-498: The Tg values of the copolymers, prepared via RAFT, were 496 measured by Differential Scanning Calorimetry….”This segment is not necessary as it is described in various sections of the manuscript. It is very repetitive.”

Line 499-500: the same recommendation as above.

In the references part: many of the references are shaded. Please homogenize according to the style of this Journal.

There are style details in various sections of the manuscript.

All the information where temperature is provided must be added a space between the number and the symbol. Review the entire manuscript.

60 °C instead 60°C

Please remove the capital letters. Few examples are shown below, but please correct throughout the manuscript.

Line 21: Please remove the capital letters: differential scanning calorimetry (DSC)

Line 44: reversible addition fragmentation chain transfer (RAFT) process

Line 49: monomer-chain transfer agent (CTA)

Line 114: polystyrene

Others minor details:

Line 100: polymerization time was 40´ (min). Please correct.

Author Response

Reviewer #1

In this work, the authors synthesized statistical copolymers of N-vinylpyrrolidone with isobornyl methacrylate by free radical and RAFT polymerization. The reactivity ratios were estimated using several methods. Also, the thermal properties of the copolymers were studied.

In my opinion there are many stylistic details that need to be corrected throughout the manuscript before being accepted. Improve the quality of images, figures, equations, etc.

We did our best to improve the quality of the manuscript following the Reviewer’s suggestions.

My comments and questions:

In the abstract part: it is recommended to include values of monomer reactivity ratios obtained in this work.

The values of the reactivity ratios were included in the Abstract.

The introductory part should be extended and enriched, for example include information on the methods to calculate the comonomer reactivity ratios. Moreover, the NVP belong to less activated monomers. Please include this information.

We have revised the Introduction section adding the information requested by the Reviewer.

In the experimental part:

The experimental conditions for the synthesis of PNVP-co-PIBMA statistical copolymers is not complete.

For example, the initial concentration of the xanthate agent and the initiator must be provided for each concentration.

We are sorry for the missing details. The requested data were added in the Supporting Information Section.

Detailed information on free radical polymerization should also be provided. For example, “the polymers were dissolved in CHCl3…”.. Do the authors refer to the polymers synthesized via free radicals? Please clarify.

All the copolymers, prepared either via free radical or RAFT copolymerization were purified by the same way, i.e., by successive dissolutions in CHCl3 and precipitations in cold methanol. This point was clarified in the text.

Line 105. The purification process of the polymers obtained must be differentiated depending on the type of polymerization.

There was no reason to employ different purification processes for the copolymers. Almost all samples were rich in isobornyl methacrylate, and therefore, easily precipitated in methanol. The monomers are readily soluble in methanol. Consequently, repetitive precipitations of the copolymers in methanol were an efficient procedure to eliminate the excess of the monomers, as was proven by the NMR spectra. Only the sample F80-20 had higher composition in PNVP. However, using cold methanol was sufficient for the purification of the copolymer. The high molecular weight of the sample facilitates the precipitation in cold methanol.

Line 149: In the 1H NMR spectrum, hydrogens of the copolymers should be assigned in detail.

Initially, we decided just to point out the hydrogen atoms of the two monomer units used for the composition analysis. However, as the reviewer suggested the complete analysis was provided in the text.

Also, a 1H NMR spectrum for R60/40 should be included. The signals of the hydrogens from the CTA will be observed in low field.

The 1H NMR spectrum for R60/40 was included in the Supporting Information Section to avoid adding more figures to the main text of the manuscript. The signals of the hydrogens from the CTA are not visible in the NMR spectrum due to the very high molecular weights of the copolymers. The resolution is not so high for this very low concentration of the samples’ end groups.

Line 241: The quality of the image in which the equations are displayed must be improved. for example: eq 13 to 15. Also eq. 18, 19.

We are truly sorry for this. The equations were re-written and their quality was substantially improved.

Line 497-498: The Tg values of the copolymers, prepared via RAFT, were 496 measured by Differential Scanning Calorimetry….”This segment is not necessary as it is described in various sections of the manuscript. It is very repetitive.”

This sentence was included in the Conclusion section. Everything reported there has been previously reported to the main text. From this point of view this statement is not a repetition. However, since the Reviewer commented on that the sentence was deleted.

Line 499-500: the same recommendation as above.

The sentence was deleted as well.

In the references part: many of the references are shaded. Please homogenize according to the style of this Journal.

 We tried to homogenize all references according to the journal’s style.

There are style details in various sections of the manuscript.

All the information where temperature is provided must be added a space between the number and the symbol. Review the entire manuscript.

60 °C instead 60°C

Please remove the capital letters. Few examples are shown below, but please correct throughout the manuscript.

Line 21: Please remove the capital letters: differential scanning calorimetry (DSC)

Line 44: reversible addition fragmentation chain transfer (RAFT) process

Line 49: monomer-chain transfer agent (CTA)

Line 114: polystyrene

Others minor details:

Line 100: polymerization time was 40´ (min). Please correct.

Al the proposed changes were corrected in the text.

Reviewer 2 Report

Pitsikalis and coworkers reported the free radical copolymerization of N-vinylpyrrolidone and isobornyl methacrylate with or without RAFT agent. It is as expected that RAFT agent assisted the polymerization in getting polymers with narrow molecular weight distribution. 

  1. The authors are suggested to briefly discuss about the difference between vinyl and acrylate monomers during polymerization, at least in terms of reactivity and how the structural difference affect polymerization.
  2. Polymerization kinetics result should be included to show the effect of RAFT agent on free radical polymerizations, with X-axis as time and Y-axis as the Mw. 
  3. Figure 2, the unique proton for each monomer seems to be hard to integrate for a precise determination on the molar ratio of the monomers. Could the authors demonstrate how the integral was obtained for these two peaks?
  4. The authors should elaborate the fundamental information of FOW and Kissinger methodologies, especially explain how the data was interpreted from these linearly fitted plots. 
  5. The authors mentioned in the Conclusion that "As was expected, RAFT methodology afforded a much better control over the synthesis of the copolymer." Is there any other significant information that the authors can conclude other than this (not the experimental details after this sentence)? Please provide.

Author Response

Reviewer #2

Pitsikalis and coworkers reported the free radical copolymerization of N-vinylpyrrolidone and isobornyl methacrylate with or without RAFT agent. It is as expected that RAFT agent assisted the polymerization in getting polymers with narrow molecular weight distribution. 

  1. The authors are suggested to briefly discuss about the difference between vinyl and acrylate monomers during polymerization, at least in terms of reactivity and how the structural difference affect polymerization.

The suitable discussion of this matter was added in the text. We agree with the Reviewer that this is an important issue and it should be mentioned in the manuscript.

  1. Polymerization kinetics result should be included to show the effect of RAFT agent on free radical polymerizations, with X-axis as time and Y-axis as the Mw. 

It was not our purpose to provide detailed kinetic analysis of the polymerization of isobornyl methacrylate via RAFT. Relative details for NVP polymerization have been reported earlier in the literature. For the study of the statistical copolymerization, we need to stop the copolymerization in rather low reaction yields in order to extract data for the calculation of the reactivity ratios. Therefore, we have not performed detailed kinetic experiments.

  1. Figure 2, the unique proton for each monomer seems to be hard to integrate for a precise determination on the molar ratio of the monomers. Could the authors demonstrate how the integral was obtained for these two peaks?

The NMR peaks in the statistical copolymers are relatively broader than those reported for the respective homopolymers. This is due to the different chemical environment, which is produced by the various sequences of the monomer units. In other words, the resonance of the characteristic hydrogen atom of the NVP monomer unit will depend on the other monomer units surrounding it, i.e., whether this NVP unit is between other two NVP units or between an NVP and an IBMA monomer unit. Therefore, for the integration we take into consideration the broad peaks, which are shown by the green bars.

  1. The authors should elaborate the fundamental information of FOW and Kissinger methodologies, especially explain how the data was interpreted from these linearly fitted plots. 

Details about these methodologies were added in the text.

  1. The authors mentioned in the Conclusion that "As was expected, RAFT methodology afforded a much better control over the synthesis of the copolymer." Is there any other significant information that the authors can conclude other than this (not the experimental details after this sentence)? Please provide.

This statement refers to the better control over the molecular characteristics obtained by RAFT polymerization compared to free radical polymerization (lower dispersity values and better control over the molecular weights). In addition, the RAFT technique allows the synthesis of more complex macromolecular architectures. We are in progress of synthesizing block copolymers of PNVP and PIBMA. Specific quantitative data were added to the conclusion section.